METHODS AND RESOURCES

# Barcoded *Asaia* bacteria enable mosquito in vivo screens and identify novel systemic insecticides and inhibitors of malaria transmission

Angelika Sturm[1]☉, Martijn W. Vos[1]☉, Rob Henderson[1], Maarten Eldering[1], Karin M. J. Koolen[1], Avinash Sheshachalam[1], Guido Favia[2], Kirandeep Samby[3], Esperanza Herreros[3], Koen J. Dechering[1]*

1 TropIQ Health Sciences, Nijmegen, the Netherlands, 2 University of Camerino, Camerino, Italy, 3 Medicines for Malaria Venture, Geneva, Switzerland

☉ These authors contributed equally to this work.
* k.dechering@tropiq.nl

**Data Availability Statement:** All relevant data are within the paper and its Supporting Information files.

## Abstract

This work addresses the need for new chemical matter in product development for control of pest insects and vector-borne diseases. We present a barcoding strategy that enables phenotypic screens of blood-feeding insects against small molecules in microtiter plate-based arrays and apply this to discovery of novel systemic insecticides and compounds that block malaria parasite development in the mosquito vector. Encoding of the blood meals was achieved through recombinant DNA-tagged *Asaia* bacteria that successfully colonised *Aedes* and *Anopheles* mosquitoes. An arrayed screen of a collection of pesticides showed that chemical classes of avermectins, phenylpyrazoles, and neonicotinoids were enriched for compounds with systemic adulticide activity against *Anopheles*. Using a luminescent *Plasmodium falciparum* reporter strain, barcoded screens identified 48 drug-like transmission-blocking compounds from a 400-compound antimicrobial library. The approach significantly increases the throughput in phenotypic screening campaigns using adult insects and identifies novel candidate small molecules for disease control.

## Introduction

Parasites and viruses that are carried by mosquitoes cause diseases such as malaria, dengue, or yellow fever. Malaria resulted in 229 million cases, causing 409,000 deaths in 2019. The use of insecticides has had large impact on control of malaria [1]. Since World War II, the range of chemical scaffolds with insecticide activity has slowly expanded, resulting in 55 chemically distinct classes of marketed insecticides available in 2019 [2]. Concurrently, resistance to these molecules has developed at a similar rate as a result of widespread use in crop protection, community and household spraying, and impregnation of bed nets [3]. As a more targeted approach, the use of oral insecticides in drug-based vector control is considered [4]. The

**Funding:** This work was financially supported by the Bill and Melinda Gates Foundation (https://www.gatesfoundation.org) through grants OPP1067662 and OPP1118462 to KJD. The funders had no role in study design, data collection and analysis, decision to publish, or preparation of the manuscript. Under the grant conditions of the Foundation, a Creative Commons Attribution 4.0 Generic License has already been assigned to the Author Accepted Manuscript version.

**Competing interests:** I have read the journal's policy and the authors of this manuscript have the following competing interests: KJD holds stock in TropIQ Health Sciences.

**Abbreviations:** DMSO, dimethylsulfoxide; MFI, median fluorescence intensity; MMV, Medicines for Malaria Venture; RLU, relative light units; R&D, research and development; SMFA, standard membrane feeding assay; TCP, target candidate profile.

endectocide ivermectin is used as an oral helminticide but also shows systemic adulticide activity against *Anopheles* mosquitoes [5]. It has shown promise as a drug that, following repeat mass drug administration to a human population at risk, reduces malaria burden by directly blocking onward pathogen transmission through reduction of the life span of blood-feeding mosquitoes [6]. Ivermectin is relatively rapidly eliminated from the blood circulation in humans, whereas modelling suggest that the duration of the mosquitocidal activity strongly drives impact of drug-based vector control [7]. Therefore, long-acting drug substances and formulations are being pursued [8,9].

As an alternative to use of insecticides for control of vector-borne disease, strategies aimed at biological control of the pathogen stages that underlie spread of the disease are emerging. These approaches have the advantage of a low risk on development of resistance. Arboviruses like Zika and dengue and protozoa such as *Leishmania*, *Plasmodium*, and *Trypanosoma* face a population bottleneck in the insect vector [10,11], with a low number of replication cycles and, hence, a low rate of accumulation of resistance mutations. In the context of malaria elimination, drug interventions targeting the transmission stages of the malaria parasite are explored [12]. Such compounds may kill or sterilise sexual stage parasites that infect mosquitoes [13,14]. Historically, antimalarial compounds have been selected on their ability to clear asexual blood stage parasitaemia that underlies clinical disease, and many of these compounds do not block transmission. More recently, compounds have emerged with a transmission-blocking component in their activity spectrum, although in many cases, this activity is not as potent as their activity against asexual blood stages [15,16]. Therefore, there is a need for novel chemical starting points for the development of malaria transmission-blocking drugs.

The requirements for drug candidates that block malaria transmission by killing the mosquito vector or by targeting the sexual stage parasites are outlined in target candidate profiles (TCP) 5 and 6, as put forward by the Medicines for Malaria Venture (MMV) [17]. These TCPs are stimulating and guiding global drug discovery efforts [18]. In the absence of a large array of validated molecular targets, these efforts rely on phenotypic screens that have a relative low throughput and, hence, generate low numbers of chemically diverse starting points [2,19,20]. In pesticide discovery, miniaturised assays in 96-well assays containing larvae are used to predict systemic activity against adult insects [21,22]. Discovery of molecules that block transmission of malaria ultimately relies on laborious membrane feeder experiments that use one container of mosquitoes for each test condition [23]. An increase in throughput of these technologies would accelerate the development of novel malaria interventions.

Previous work has demonstrated the use of phage display technology to enable large-scale screens for peptides that block *Plasmodium* development in the mosquito vector [24]. Killeen and colleagues showed the feasibility of feeding mosquitoes on arrays of blood meals encoded with scFv-displaying phages that could be recovered, identified, propagated, and retested from individual mosquitoes [25]. This introduced the concept of large-scale screens, where active substances could be identified by enrichment of their cognate phage barcode in mosquitoes with the phenotype of interest. Here, we build on this concept and present a technique that significantly improves the throughput of compound testing in phenotypic assays using adult mosquitoes. It allows screening of multiple molecules using barcoded blood meals in multi-sample arrays. We used a genetically engineered prokaryotic symbiont, α-Proteobacteria of the genus *Asaia*, which stably associate with a number of sugar feeding insects [26]. Upon ingestion with a glucose or a blood meal, *Asaia* actively colonises the insect midgut within 1 or 2 days and spreads from there to most other organs [27,28]. We transformed *Asaia* strains with plasmids that carry individual short DNA barcodes. Following feeding of mosquitoes on arrays of blood meals with test compounds, these DNA barcodes were recovered from the mosquito in order to deconvolute the feeding pattern and identify active compounds. We used

this technique to identify systemic insecticides and malaria transmission-blocking compounds from libraries of small molecules.

## Results

### Membrane-feeding mosquitoes with a barcoded blood meal

We envisaged to use DNA-encoded blood meals in 96-well plates presented to hematophagous insects, allowing deconvolution of the feeding pattern. A previous study has shown the feasibility of feeding *Anopheles* mosquitoes on phage-encoded 96-well plates covered with Parafilm membrane [25]. We developed a device to evenly stretch a Parafilm membrane in 2 directions (S1A Fig). A hydraulic press was used to push the stretched membrane firmly down on a 96-well microtiter plate containing prewarmed blood meals (S1B Fig). The plate was placed upside down on a cage of mosquitoes and warmed with a preheated aluminium heat block that was routed to exactly fit the base of the microtiter plate (S1C Fig). Feeding efficiency depended on the temperature of the heat block. At 45°C, feeding performance was consistently above 90%, which was comparable or even better than a method using conventional glass feeders (Fig 1A). Video analysis of feeding behaviour on a cage with approximately 300 mosquitoes suggested sampling of all wells across the plate (S1 Movie).

Initial experiments with DNA-encoded blood meals using phosphorothioate oligonucleotides were unsuccessful, as the DNA oligo's were rapidly cleared by the mosquito and would not allow evaluation of phenotypes that take several days to develop (S2A Fig). For more stable introduction of a DNA tag, we evaluated 2 potential bacterial carriers, the midgut symbionts *Pantoea agglomerans* and *Asaia SF2.1* [27,30]. *Pantoea* showed a strong effect on transmission of *Plasmodium falciparum* malaria parasites to *Anopheles stephensi* mosquitoes (S2B and S2C Fig). This was not observed for *Asaia* (S2D and S2E Fig), and all subsequent experiments used *Asaia* SF2.1. We generated a collection of 50 bacterial stocks each with a unique DNA tag (S1–S3 Tables). Pilot experiments with *Aedes aegypti* mosquitoes fed on a grid with 2 differently barcoded blood meals each placed in 3 different wells in a 96-well plate showed that 2 days after feeding, 100% of the fed mosquitoes was successfully tagged with a single barcode (Fig 1B). Out of these, 74% contained barcode 1, and 26% contained barcode 2. This uneven distribution of barcodes may relate to the relatively small sample size. Analyses of a different cohort of mosquitoes 8 days after feeding showed a more even distribution, with equal proportions (36%) of mosquitoes having a single barcode (Fig 1C). At this time point, 27% showed a signal for both barcodes. This may result from cross-feeding, although cross-feeding was not observed for the cohort analysed 2 days postfeeding. Alternatively, cross-contamination may occur later on in the experiment, possibly through contact with mosquito diuresis fluids, excrements, or the cotton pad that was used for glucose feeding during the experiment. Pilot experiments with *A. stephensi* mosquitoes showed similar results, with roughly equal proportions of mosquitoes with a single barcode and 20% of mosquitoes with 2 barcodes (Fig 1D). To prevent cross-contamination of barcodes in subsequent experiments, we limited exposure to glucose pads to 2 hours per day while changing pads daily. In addition, mosquitoes were transferred to new cages directly after feeding to reduce exposure to diuresis fluid on the cage floor.

### Phenotypic screen for systemic insecticide activity

Based on these initial pilot experiments, we devised a strategy for multiplex detection of barcode signals in order to enable larger phenotypic screens (Fig 1E). We explored suitability of this screening principle for phenotypic screening for systemic insecticides using fipronil as a reference compound. *A. stephensi* mosquitoes were fed on a 96-well plate with 24 barcoded blood meals, half of them containing 10 μM fipronil and the other half 0.1% DMSO as a vehicle

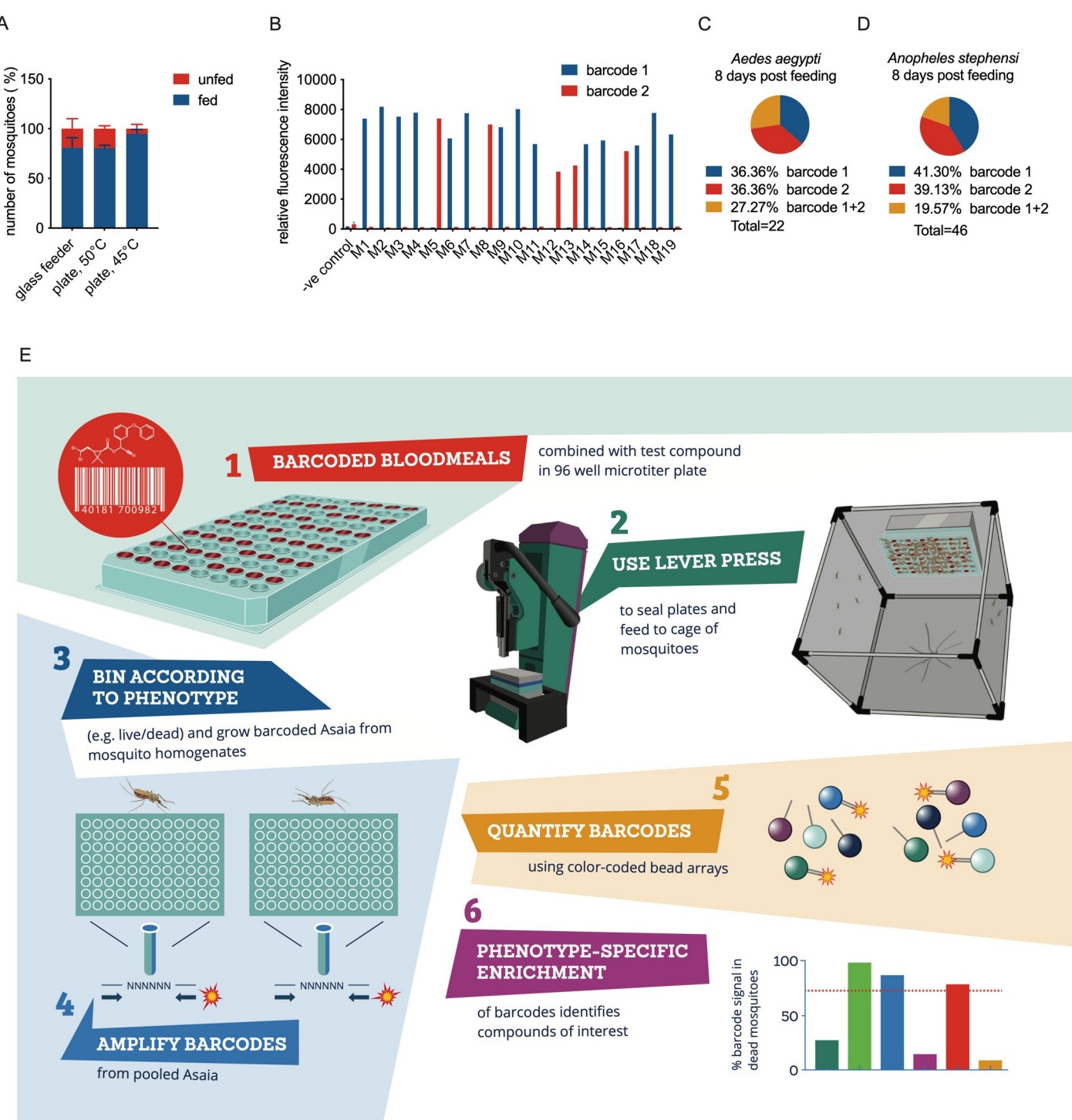

**Fig 1. Tagging mosquitoes with a molecular barcode through feeding on microtiter plates.** (A) Comparison of feeding efficiency of *Anopheles stephensi* mosquitoes between glass feeders and microtiter plates. An aluminium block heated to 50˚C or 45˚C as indicated on the x-axis was placed on top of the plate. The figure shows the percentage of fed and unfed mosquitoes in the cage. (B) Barcode signals in individual *Aedes aegypti* mosquitoes fed on a grid containing 2 different barcodes. The mosquitoes were analysed 2 days after feeding, and the figure shows relative fluorescence intensities for the 2 barcodes in individual mosquito samples. (C) Prevalence of barcode positive *A. aegypti* mosquitoes 8 days after feeding on a grid of 2 distinct barcodes. (D) Prevalence of barcode positive *A. stephensi* mosquitoes 8 days after feeding on a grid of 2 distinct barcodes. (E) Outline of screening strategy for identification of systemic insecticides. Mosquitoes were fed on microtiter plates containing barcoded blood meals supplemented with test compounds. Two days after blood feeding, mosquitoes were split into pools of live and dead mosquitoes, and *Asaia* bacteria were grown from homogenates of individual mosquitoes in 96-well liquid cultures under kanamycin selection pressure. Barcodes were then amplified by PCR using a fluorescently labelled primer pair that binds a common sequence flanking the DNA barcode sequence. Following amplification,

barcodes were quantified by multianalyte profiling using DNA oligos coupled to colour-coded microspheres [29], which resulted in a fluorescence signal for each barcode depending on the quantity of the barcode in the PCR amplification product. Barcodes enriched in the dead mosquitoes identified compounds with systemic adulticidal activity, whereas detection of barcode signals from the live mosquitoes were used to verify sampling of barcodes that were missing in the pool of dead mosquitoes. Underlying data can be found in S1 Data.

control (0.1% DMSO). Moreover, 48 hours after feeding, we retrieved all blood-fed mosquitoes from the cage. Of these, 70 of were alive, and 77 were dead. Analyses of barcode presence in individual mosquitoes showed that 100% of the mosquitoes were successfully tagged with a barcode. Of these, 124 (84.3%) showed a single barcode, 21 (14.3%) showed 2 barcodes, and 2 mosquitoes (1.4%) showed 3 barcodes (Fig 2A).

The barcodes associated with DMSO and fipronil segregated with a live and death phenotype, respectively. In the live cohort, 2 mosquitoes were positive for both DMSO and fipronil associated barcodes (Fig 2A). This may be explained by intake of a sublethal quantity of fipronil or cross-contamination of barcodes postfeeding. In the dead cohort, all mosquitoes except for one were found positive with a fipronil-associated barcode. Deconvolution of the feeding pattern showed that wells were sampled on average by 7 mosquitoes, with a range of 3 to 17 mosquitoes (Fig 2B). Cross-contamination of barcodes, either by uptake of multiple blood meals or postfeeding contact with barcode-containing material, was low across the plate. Since every well was sampled by multiple mosquitoes, the contaminating signal may only make up a small contribution to the summed signals from that well. For each barcode, we calculated the total signal from all mosquitoes positive a particular barcode in live or dead mosquitoes. The results show a clear compound and phenotype-dependent enrichment of barcodes (S3 Fig) and suggest that analyses of pooled samples may correctly annotate activity of a test compound in a barcoded blood meal. To test this experimentally, we created pools of *Asaia* bacteria rescued from live and dead mosquitoes, respectively, and analysed barcode intensities per pool in a single multiplex reaction. Barcodes in DMSO control meals were predominantly found in live mosquitoes, whereas barcodes in fipronil-containing blood meals were associated with the dead phenotype (Fig 2C). The highest contaminated signal was observed with barcode 27 that was associated with a DMSO containing blood meal but showed 28% of the total signal originating from dead mosquitoes (Fig 2C). This barcode was located in well A9 and was retrieved from a total of 7 mosquitoes, of which 1 mosquito was dead at the time of sampling (Fig 2B). The combined data highlight the feasibility of multiplex barcode detection in pools of mosquitoes binned according to the phenotype of interest.

## Screening pesticides against *Anopheles stephensi* mosquitoes

The above experiments demonstrated the feasibility of multiplex barcode detection in pools of mosquitoes binned according to the phenotype of interest. Using this strategy, we screened a collection of 83 chemically diverse pesticides to identify novel candidates for drug-based vector control approaches. Compounds were initially tested at 1 μM in duplicate with up to 48 samples per plate (S4 Fig). For each phenotype (live or dead mosquitoes), the *Asaia* cultures were pooled, and DNA barcodes from each pool were amplified and quantified. All compounds with ≥50% of the barcode signal in the dead mosquitoes were subsequently tested at 100 nM, whereas all inactive compounds (<50%) were tested at 10 μM concentration. From a total of 188 experimental conditions in 4 feeding experiments, a total of 2,727 mosquitoes were analysed. Of these, 952 were dead 48 hours after feeding. Moreover, 3 wells were not sampled. Compounds MMV03891 and MMV1577456 were not sampled in the initial run when tested at 1 μM but showed a barcode signal when tested at the same concentration in a repeat experiment, suggesting the initial lack of sampling was not due to interference, e.g., through a

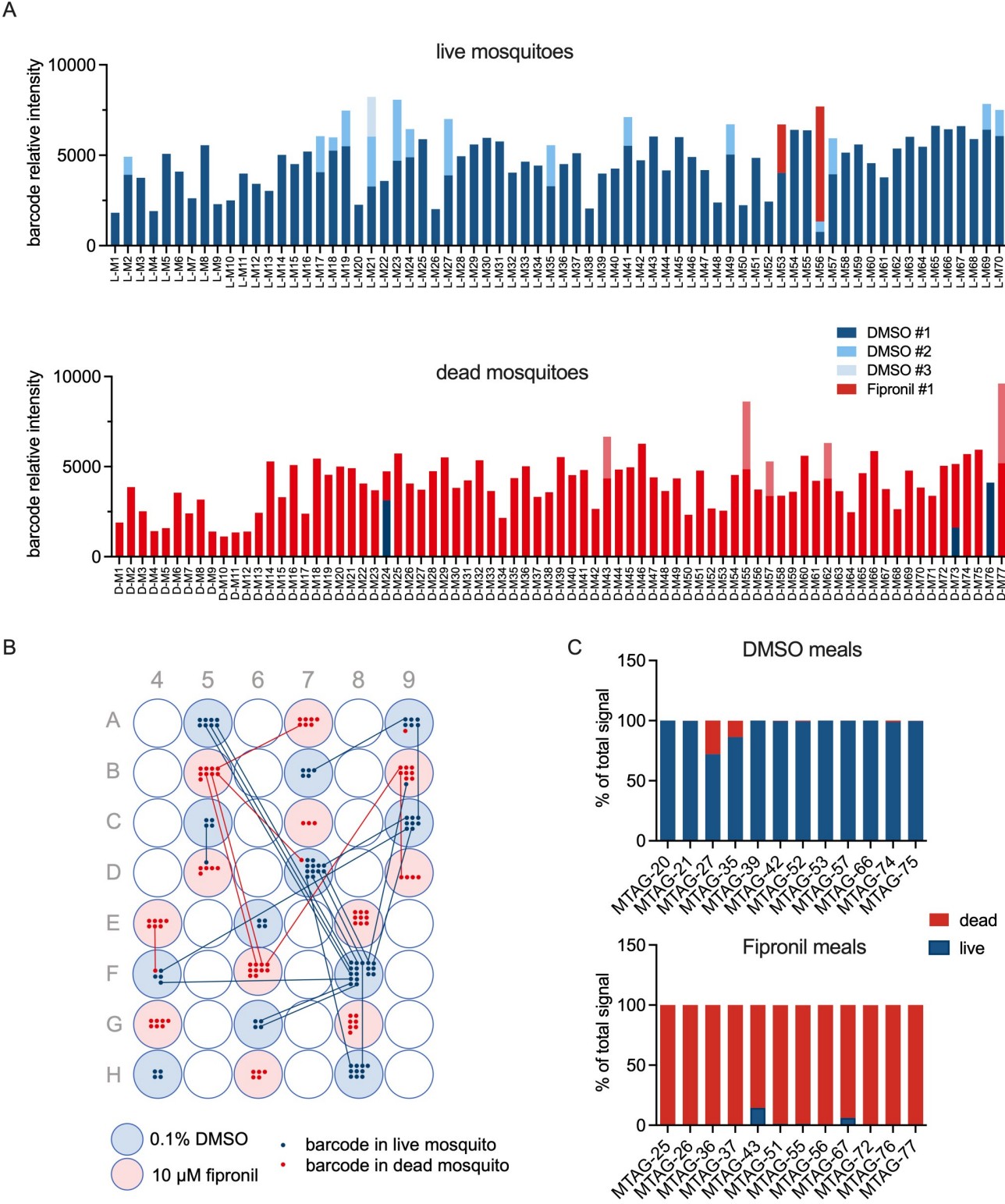

**Fig 2. Proof of principle for systemic insecticide screen.** *A. stephensi* mosquitoes were fed on a grid of 12 individually barcoded vehicle control (0.1% DMSO) and 12 individually barcoded insecticide (10 μM fipronil) blood meals. Two days after feeding, phenotype (live/dead) and barcode presence was determined for individual mosquitoes. **(A)** Barcode signals in the live (upper panel) and dead (lower panel) cohort of mosquitoes. The figure shows the sum of barcode signals in individual mosquitoes. For mosquitoes showing multiple signals, barcode signals were grouped according to origin (DMSO or fipronil) and ranked on signal strength. DMSO #1, #2, and #3 indicates the highest, second highest, and third highest signal, respectively, originating

from a DMSO well. There were no mosquitoes with more than 3 barcode signals above background. For the fipronil barcodes, we did not observe any mosquito with more than 2 barcodes originating from a fipronil-containing well. **(B)** Deconvolution of feeding/cross-contamination pattern. Blue and red shading indicates wells that contained a DMSO or a fipronil blood meal, respectively. Dots indicate mosquitoes found positive for a barcode originating from that well, with blue dots indicating the mosquito was alive, whereas red dots indicate dead mosquitoes. Lines indicate mosquitoes that were found positive for more than 1 barcode. **(C)** Analyses of pooled samples. *Asaia* rescued from the mosquito midguts were binned according to the mosquito phenotype (live or dead), and barcodes were amplified from the pooled samples. The graphs show the proportion of barcode signal originating from live or dead mosquitoes for DMSO (upper panel) or fipronil barcoded blood meals. Barcodes indicated on the x-axis are listed in S3 Table. Underlying data can be found in S1 Data.

gustatory effect preventing blood feeding or antimicrobial action against the barcoded *Asaia*. Compound MMV1633827 was sampled when tested at 1 μM but not at 10 μM. The latter concentration was not repeated, and we cannot exclude that this compound interfered at some point in the process. Fig 3A shows the barcode signals for the negative (DMSO) and positive (deltamethrin and fipronil) control wells. The data indicate a clear treatment-dependent distribution of barcode signals over the 2 phenotypes, with an average of 100% of the signal in the dead mosquitoes for barcodes associated with either one of the insecticides and 0% for barcodes associated with the DMSO control wells. Individual values below 0% or above 100% arise from the background subtraction in the data analyses, as in some instances background values in unfed mosquitoes were slightly higher than in negative mosquitoes from the experiment (S1 Data). The barcode distribution for all 189 experimental conditions showed a similar pattern, with a subpopulation around 0% and another around 100% associated with the death phenotype (Fig 3B). Fig 3C and S1 Data show the data for individual compounds. For 4 compounds, we tested 2 different chemical batches, listed under separate MMV batch codes. Of these, methomyl (MM003972-04 and MM003972-05), nitempyram (MMV673126-3 and MMV673126-4), and amitraz (MMV002471-05 and MMV002471-06) showed consistent results between the 2 batches. For rotenone, batch MMV002519-09 did show activity at 10 μM, whereas batch MMV002519-11 did not. Compounds from the class of avermectins appeared to be among the most active compounds with more than 90% of the barcode signal associated with the death phenotype at test concentrations of 100 nM and 1 μM (Figs 3C and S5). Likewise, phenypyrazoles fipronil and vaniliprole and the isoxazoline fluralaner showed potent killing activity. Other phenylpyrazoles showed less potent activity, with more than 95% of the signal in the dead pool of mosquitoes when tested at 1 μM but not at 100 nM. The class of neonicotinoids was also enriched among the set of active compounds, with 70% to 100% of the barcode signal associated with the death phenotype when tested at either 1 or 10 μM. To validate the results from the barcoded screen, we tested a number of compounds in traditional glass feeder membrane feeding experiments. These experiments confirmed systemic insecticide activity for all compounds tested (fipronil, deltamethrin, chlorfenapyr, abamectin, fluralaner, vaniliprole, and spinetoram), with $IC_{50}$ values ranging from 3 nM for abamectin to 3,173 nM for chlorfenapyr (Fig 3D).

## Compound screen for transmission blockade of *Plasmodium falciparum*

We developed a method for screening for inhibition of pathogen transmission, using the human malaria parasite as a model organism. In the procedure, outlined in S6 Fig, a transgenic *P. falciparum* reporter strain was used to infect *A. stephensi* mosquitoes by feeding on arrays of barcoded blood meals containing test compounds. This reporter produces a clear luminescence signal that is linearly correlated with the number of oocysts in the mosquito midgut as described previously [31,32]. Eight days after blood feeding, mosquito infection status was assessed by luminescence measurement and *Asaia* bacteria were then rescued from individual mosquitoes and pooled into separate bins for infected and uninfected mosquitoes. Enrichment

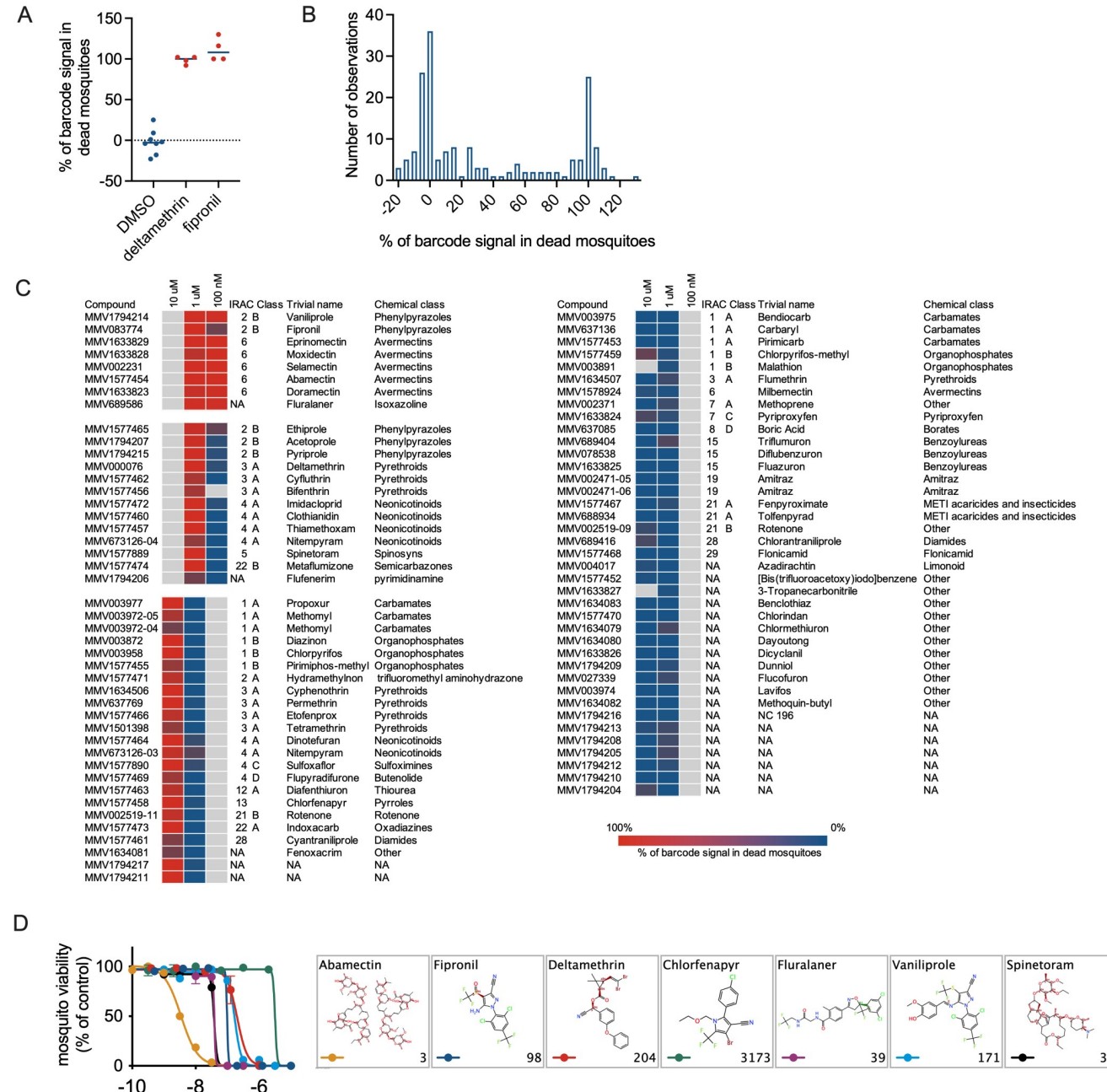

**Fig 3. Screening of a collection of pesticides/insecticides. (A)** Assay controls. The figure shows the percentage of the total barcode signal that associated with the death phenotype for barcodes in control wells containing vehicle (0.1% DMSO), 10 μM deltamethrin or 10 μM fipronil. **(B)** Distribution of barcode enrichment for all test conditions and barcodes. The figure shows a histogram of the proportion of the signal that was retrieved from dead mosquitoes relative to the total signal (dead plus live mosquitoes) for a particular barcode. **(C)** Heatmap of barcode enrichment in dead mosquitoes for the compounds and test concentrations indicated. Compounds were initially tested at 1 μM. Inactive compounds were then tested at 10 μM, whereas active compounds were tested at 100 nM. The colour shading indicates the percentage of barcode enrichment in the dead mosquitoes. Grey colours indicate conditions that were not tested/sampled. **(D)** Confirmation of systemic insecticide activity through traditional membrane feeding experiments using glass feeders. The compounds indicated in the legend were tested at multiple concentrations in duplicate. Error bars indicate standard deviations. IC50 estimates (in nM) from nonlinear regression analysis are indicated in the lower right corners of the panels depicting the compound structures. Underlying data can be found in S1 Data.

of barcode signals in the pool of uninfected mosquitoes identified wells containing a transmission-blocking test specimen. We screened the open access Pathogen Box, a collection of 400 chemically diverse and drug-like molecules selected for their potential action against a variety of pathogens underlying tropical infectious diseases (S7 Fig) [33]. Compounds were preincubated with stage V gametocytes for 24 hours prior to mosquito feeding, in order to identify compounds with a gametocytocidal mode of action and in line with TCP5 of the MMV [17]. The total experiment involved 441 barcoded samples that were processed in 9 batches involving analyses of 4,545 mosquitoes. Of these, 1,794 showed a luminescence signal within 3 standard deviations of average background signal from unfed control mosquitoes and were considered uninfected (Fig 4A). All barcodes were successfully detected in either uninfected, infected, or both mosquito pools. For barcodes associated with atovaquone, on average, 96% of the barcode signal was retrieved from the uninfected pool of mosquitoes (Fig 4B). For the DMSO controls wells, the percentages of the barcode signals in the uninfected mosquitoes relative to the total barcode signals averaged at 18%. This is in line with the experimental variation in mosquito infection success rates [15,34]. Subsequently, we arbitrarily set the threshold for transmission-reducing activity at 80% of the barcode signal in the uninfected pool of mosquitoes, which separates the atovaquone from the DMSO vehicle controls with one exception (Fig 4B). From the collection of 400 Pathogen Box compounds, 48 compounds met this criterion (Fig 4C, S1 Data). To verify this result, we selected 21 chemically diverse compounds for which barcodes were enriched in uninfected mosquitoes and tested these in individual membrane feeding experiments using regular glass feeders. Of these, 19 compounds reduced oocyst intensities by 80% or more in the glass feeder experiments, indicating a low false positive rate in the barcoded assay (Figs 4D and S8).

To gain insight in the range in potencies of the transmission-blocking hit compounds, we randomly selected 5 compounds for full dose–response analyses in glass feeder experiments. Compounds MMV1088520, MMV667494, and MMV022029 originate from the malaria compound set, and the last 2 compounds were previously annotated as gametocytocidal (S4 Table). In the dose–response analyses, $IC_{50}$s were determined at 1,078, 18, and 56 nM, respectively (Fig 4E). MMV688122 originates from a *Mycobacterium* screen and blocked transmission with an $IC_{50}$ of 1 μM, whereas MMV675968 is part of the *Cryptosporidium* collection of the pathogen box and showed an $IC_{50}$ of 406 nM. The combined results indicate that the barcoding technology significantly increases throughput in membrane feeding assays and leads to identification of novel chemical starting points for control of malaria.

## Discussion

Conventional testing of the effectiveness of substances on longevity or vector capacity of live insects is labour intense and mostly allows only for a small number of molecules to be tested simultaneously. We have developed a technique that improves the throughput of compound testing in order to fuel pipelines for discovery of pesticides and disease transmission-blocking drugs. To do this, we had to overcome 3 distinct technical challenges: feeding mosquitoes on multiwell plates, tagging blood-fed mosquitoes with a unique well identification code, and multiplex detection of these identification codes. We used a custom designed parafilm membrane stretcher in combination with a hydraulic press to firmly seal 96-well plates filled with blood meals. The plate feeding method proved just as effective as conventional glass feeders. In order to tag mosquitoes stably throughout the course of the experiment, we used the insect midgut symbiont *Asaia* strain SF2.1, transformed with DNA barcoded plasmids. In line with published data [26,35], we observed efficient colonisation of *Anopheles* and *Aedes* mosquitoes when *Asaia* bacteria were included with the blood meal. Previously, Killeen and colleagues

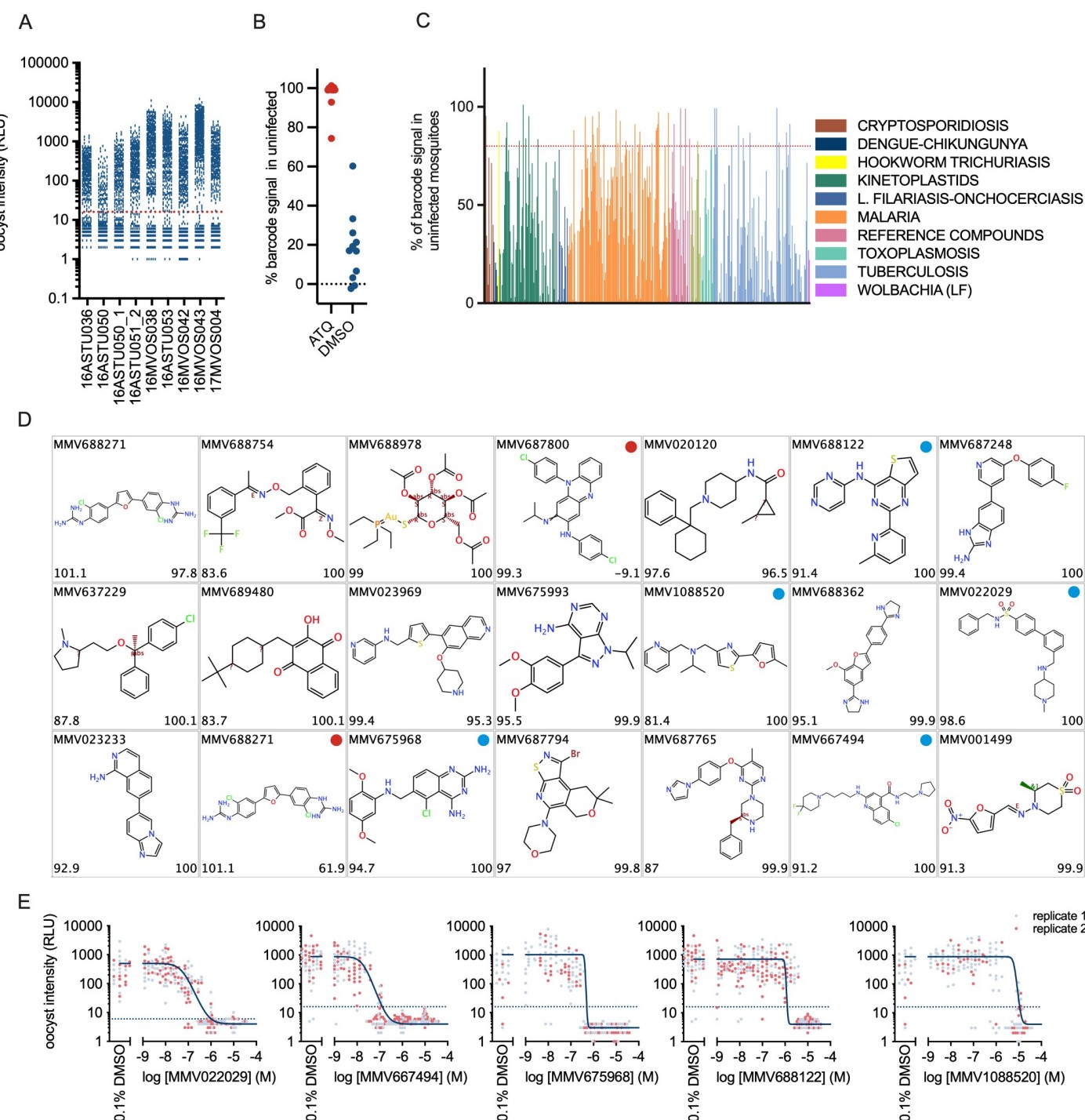

**Fig 4. Identification of malaria transmission-blocking compounds.** The open-source MMV Pathogen Box was screened in a barcoded assay for *P. falciparum* transmission using a luminescent reporter parasite. Stage V gametocytes were preincubated with test compound at 10 or 20 μM as indicated in S1 Data and fed to *A. stephensi* mosquitoes through an *Asaia* barcoded blood meal. Eight days after feeding, infection status was determined by a luminescence assay. Barcodes were retrieved from infected (luciferase positive) and uninfected (luciferase negative) mosquitoes and quantified. **(A)** Oocyst intensities in mosquitoes from 9 experimental runs that were used to screen a collection of 400 compounds. The figure shows luminescence activities in individual mosquitoes. The red dotted line indicates the threshold (average background + 5σ) that was used to discriminate infected from uninfected mosquitoes. **(B)** Assay controls. The figure shows the percentage of the total barcode signal that associated with the uninfected phenotype for barcodes in control wells containing vehicle (0.1% DMSO) or 10 μM atovaquone. **(C)** Proportion of barcode in the uninfected mosquitoes for all compounds tested. Colours indicate the origin of the compound sets that compose the Pathogen Box. The red dotted line indicates the threshold for selection of active compounds (≥80% of the total barcode signal derived from uninfected mosquitoes). **(D)** Compounds selected for confirmation experiments. The lower left corner of each panel indicates the proportion of barcode signal that was retrieved from uninfected mosquitoes.

The lower right corner indicates the percentage reduction in oocyst intensity that was observed in an SMFA. All compounds except for 2 compounds indicated with a red dot showed transmission-blocking activity in the SMFA. Compounds identified by a blue dot were selected for full dose–response analysis. **(E)** Dose–response analysis in SMFA for selected compounds. All test concentrations were analysed in duplicate in replicate feeders. For each feeder, infection status of individual mosquitoes was analysed through luminescence analysis. The figure shows oocyst intensities expressed as relative light units for individual mosquitoes. The solid lines indicate the fitted dose–response curves. Dashed lines indicate the luminescence background level in uninfected mosquitoes. Underlying data can be found in S1 Data.

introduced a phenotypic screening concept based on phagemid encoded multisample arrays [25]. This approach led to 95% of successfully tagged mosquitoes with the marker lasting for 3 days only. In capture and release experiments, ectopic DNA oligonucleotides have been used to stably tag mosquitoes during their entire life span [36]. Our data indicate that upon ingestion with the blood meal, DNA oligonucleotides are rapidly eliminated from the mosquito in spite of their nuclease-resistant phosphorothioate backbone. By contrast, ingested *Asaia* symbionts stay with the mosquito for life and thereby makes long-term applications possible [27]. Sampling intensity of *Asaia* encoded blood meals differed between wells, and some wells were sampled by more than 1 mosquito. This is in line with the biting behaviour observed with the Vectorchip that contains arrayed glucose meals to sample saliva from individual mosquito bites [37]. By using a surplus of mosquitoes relative to the number of blood meals, we ensured that every well was sampled by multiple mosquitoes. Small contaminations with barcodes from cross-feeding mosquitoes did not impact overall barcode enrichment in pooled analyses. In these analyses, the number of PCR reactions needed to detect individual barcodes was significantly reduced by using common amplification regions and multiplex detection of barcode sequences.

Our screen of a collection of pesticides exemplifies the application of the barcoding technology for discovery of novel systemic insecticides. Compounds like fluralaner and nitempyram are used as oral drugs for tick and flea control in veterinary medicine [38,39] and led to enrichment of barcodes in the dead population of mosquitoes. In addition, a number of phenylpyrazoles emerged as hits with blood-borne mosquitocidal activity against *Anopheles*. Fipronil shows a very long half-life in mammalian circulation [40] and was shown to have potent and long-lasting mosquitocidal effects when administered to cattle [41]. For other compounds from the phenylpyrazole class the systemic insecticide activity in a blood meal is less well documented, but our data show that these molecules show promise for drug-based vector control, provided they show an excellent safety profile in human. Based on the reported mammalian long in vivo half-life of fluralaner [42], this compound was selected as a promising candidate for drug-based vector control and analysed in further detail. The results, which are described elsewhere [9], showed potent killing activity against a wide range of vector species at concentrations that are in line with drug levels predicted to circulate for several months following a single human oral dose.

In order to exemplify a screen for vector-borne pathogen transmission, we used the barcoding technology to identify compounds that block *Plasmodium* development in *Anopheles* mosquitoes. Using the Pathogen Box collection and a selection criterion of ≥80% barcode enrichment in uninfected mosquitoes, we observed an overall hit rate of 12%. This relatively high hit rate may be explained by a biased composition of the pathogen box towards pharmacologically active compounds. A subset of 125 compounds from this collection is annotated as malaria hit compounds, as they showed $IC_{50}$s of 2.1 μM or better against *P. falciparum* Dd2 asexual bloodstage parasites (https://www.mmv.org/mmv-open/pathogen-box). Out of these 125, 23 (18%) appear to block transmission in the barcoded screen, which is a higher number than the one predicted on basis of gametocyte viability assays [43]. This is conceivable, as the in vivo transmission assay captures a wide range of potential mode of actions, including ones

that incapacitate gametocytes by nonlethal ways, e.g., by prevention of gamete formation or sterilisation of resulting gametes [44]. Hit rates were 9% and 10% for compounds originating from tuberculosis and kinetoplastid hit collections that were well represented in the Pathogen Box with 116 and 70 compounds, respectively. This illustrates the strength of cross-screening bioactive molecules against a large panel of pathogen species. This notion is in line with previous observations that libraries of small molecules preselected for activity against one protozoan parasite showed high hit rates against a wider variety of pathogens [45–47]. MMV675968 identified here as a *P. falciparum* transmission-blocking molecule belongs to a class of dihydrofolate reductase inhibitors with activity against a range of protozoa and was recently shown to block growth of *Acinetobacter baumannii* [48,49]. In theory, such cross-reactivity may affect the *Asaia* bacteria used in our barcoded screening strategy. As our method comprehensively monitors barcode presence in all blood-fed mosquitoes, this would lead to a total absence of the barcode in either phenotype. For the 483 compounds in the combined screens presented here, we observed successful retrieval of barcode in 482 instances, indicating a relatively low hit rate against the barcode-bearing *Asaia* bacteria.

The transmission-blocking hits described here are attractive starting points for further optimisation as they obey to rule of five principles, i.e., have no more than 5 hydrogen bond donors, no more than 10 hydrogen bond acceptors, a molecular mass $<500$ g/mol and a $\log P < 5$ [50]. In addition, all compounds have in vitro and in vivo pharmacokinetic data available (https://www.mmv.org/mmv-open/pathogen-box). For example, in rat pharmacokinetic studies, hit compound MMV687248 showed 38% absorption and clearance of 12.4 ml/min/kg, which is a reasonable starting point for further pharmacological evaluation. Ultimately, these should address the ability of a drug to reduce the parasite load in a mosquito to zero, as a single oocyst that develops in the mosquito midgut can give rise to sufficient salivary gland sporozoites to transmit the disease [51].

Historically, phenotypic screening has driven drug research and development (R&D) pipelines for infectious diseases, and it has been to a larger or lesser extent been in vogue in other therapeutic areas [52]. It is attractive as it captures complex biology in the absence of a priori knowledge of molecular mechanisms of disease. Recent advances in cell biological, imaging, and data analyses techniques have brought it back in the spotlight [53]. The methods described here expand the possibilities for phenotypic live insect screens. In line with published data, we observed stable colonisation of *A. stephensi* and *A. aegypti* mosquitoes by *Asaia* bacteria [26,54]. Applications beyond the examples provided in this paper are conceivable. For example, it should be possible to identify gustatory modulators through barcodes not sampled in arrayed screens. The incubation time in the screen for systemic insecticides presented here could be extended to screen for compounds with a slow mode of action, possibly selecting compounds less prone to development of resistance. Since *Asaia* is transmitted vertically, screening for barcodes that are absent on eggs or in progeny may identify mosquito contraceptives that reduce fecundity [55]. *Asaia* has been found to associate with other sugar-feeding, phylogenetically distant genera of insects, for example, the leafhopper *Scaphoideus titanus*, the vector for Flavescence Dorée, a grapevine disease [56]. This host flexibility makes *Asaia* an attractive tool for tagging a large variety of pest insects, for the purpose of the discovery of novel molecules for pest and disease intervention.

## Materials and methods

### Pilot experiments using modified DNA oligonucleotides

In pilot experiments, mosquito blood meals were tagged with a DNA nucleotide with phosphorothioate backbone modifications to increase nuclease resistance (S1 Table). DNA was

isolated from individual fed mosquitoes using phenol/chloroform extraction directly after feeding and after 24, 72, and 144 hours postfeeding. Presence of the modified oligo in the extracted DNA samples was assessed by semiquantitative real time PCR using primers MWV 303 and MWV 304 (S1 Table) and a fluorescent TaqMan MGB probe (ThermoFisher, Breda, the Netherlands).

### Barcode construction and transformation of *Asaia* SF2.1

Plasmid pMV170 for transformation of *Asaia* was derived by amplification of a multiple cloning site from pMV-FLPe [57] with primer pair MWV 371 and MWV 374 (S1 Table) and introducing it into the NcoI/AatII sites of vector pBBR122 (Mobitec, Goettingen, Germany). Barcode sequences, compatible with detection using MAGPlex-TAG microspheres (Luminex, 's Hertogenbosch, the Netherlands) were generated by hybridisation of complementary primer pairs (S2 Table) and cloned into pMV170 using SpeI/AflII restriction digestion and ligation. Resulting plasmids were introduced into *Escherichia coli* DH5α competent cells (Thermo Fisher) by heat shock transformation, yielding a collection of 50 barcoded plasmids (S3 Table). Barcoded plasmids were next extracted from *E. coli* using the PureYield Plasmid Miniprep System (Promega, Leiden, the Netherlands) and subsequently introduced into *Asaia* sp. SF2.1 described previously [27]. For transformation, *Asaia* cells were cultured in GLY medium (25 g/litre glycerol, 10 g/litre yeast extract, pH 5), and competent cells were prepared as previously described [27]. Next, 65 μl of the competent cells were mixed with 1 μl (approximately 50 ng/μl) plasmid and electroporated using a BTX electroporation system at 2.0 kV and 186 ohm in a prechilled 1 mm cuvette. Moreover, 935-μl prechilled GLY medium was added, and bacteria were incubated at 30°C for 4 hours without antibiotic before plating on GLY agarose plates containing 100 μg/ml kanamycin. Plates were incubated at 30°C for 48 hours, and single colonies were picked and sequence verified.

### Preparation of barcoded blood meals and plate feeding

Barcoded *Asaia* bacteria were grown overnight at 30°C to early log phase ($OD_{600}$ 0.5 to 0.8) in a deep-well plate (Sarstedt, Nümbrecht, Germany) in 300-μl GLY medium supplemented with 100 μg/ml kanamycin per well. Bacteria were next diluted in heat inactivated human serum (type A) and combined with human red blood cells (type O) to achieve a final density of $10^6$ cfu/ml and a haematocrit of 50%. Microtiter plates were filled with 160 μl of blood meal per well and sealed with a membrane (Parafilm M, PM999, VWR, Amsterdam, the Netherlands) that was stretched to about 250% its original dimensions in both directions using a custom build device (S1A Fig) and applied using a lever press (S1B Fig). The plates were kept warm (37°C) and placed upside down on top of a mosquito container sealed with mosquito netting. An aluminium block routed to fit the base of the microtiter plate and preheated to 45°C was put on top to warm the plate (S1C Fig). Experiments were performed with 3- to 5-day-old females of *A. stephensi* mosquitoes (Sind-Kasur Nijmegen strain) reared at the insectary of the Radboud University Medical Center [58] or *A. aegypti* (Rockefeller strain, obtained from Bayer, Monheim, Germany) reared at Wageningen University [59]. For a plate containing 48 barcoded blood meals, we used approximately 300 mosquitoes per container, and for experiments with other sample sizes, the number of mosquitoes was adjusted proportionally. Mosquitoes were allowed to feed for 20 minutes after which the mosquitoes were maintained at 26°C and 70% to 80% humidity.

### Recovery and detection of barcode sequences

Mosquitoes were washed in 70% ethanol followed by 3 washes in PBS (137 mM NaCl, 2.7 mM KCl, 10 mM $Na_2HPO_4$, 1.8 mM $KH_2PO_4$, pH 8.0). Individual mosquitoes were transferred to

wells in shallow 96-well plates, combined with Zirconium beads and homogenised in 60-μl PBS using a Mini-Beadbeater-96 (Biospec, Bartlesville, Oklahoma, United States). A total of 15 μl of each of the mosquito homogenates was subsequently transferred to a deep-well plate containing 300-μl GLY medium supplemented with 100 μg/ml kanamycin and 2 μg/ml amphotericin B. The plates were sealed with a gas permeable breathing seal (Greiner Bio-One, Alphen aan de Rijn, the Netherlands), and *Asaia* bacteria were grown to the stationary phase by incubation at 30°C with continuous shaking (220 rpm) for at least 72 hours. In initial experiments, barcodes were amplified from individual *Asaia* cultures by PCR. In later phenotypic screening experiments, *Asaia* cultures from mosquitoes with the phenotype of interest were pooled. For comparative analyses (e.g., live versus dead mosquitoes), mock cultures with an unrelated barcode were added to make up for differences in sample sizes between the 2 pools. This to prevent differences in amplification efficiencies due to different numbers of PCR templates and, as a result, a bias in the barcode representation. Barcodes were amplified using forward primer MWV 486 and a 5'-biotinylated reverse primer MWV 358 (S1 Table) using standard PCR conditions with Gotaq G2 flexi DNA polymerase (Promega). The biotinylated PCR products were then hybridised to a pool of MagPlex-TAG microspheres (S3 Table) according to the manufacturer's instructions (Luminex) with some adaptations. Briefly, 33 μl of microsphere mixture was prepared in 1.5 X TMAC hybridisation solution (1× TMAC = 3M Tetramethyl ammonium chloride, 50 mM Tris, 1 mM EDTA, and 0.1% SDS at pH 8.0) with about 1,000 beads per barcode for all 50 barcode sequences. This was then mixed with 2 μl from the barcode amplification reactions and 15 μl TE buffer (10 mM Tris/1 mM EDTA, pH 8) and incubated for 15 at 52°C. Next, 35 μl of reporter mix was added, containing 14.3 ug/ml SAPE (Streptavidin, R-Phycoerythrin Conjugate) and 0.24% bovine serum albumin in TMAC buffer, resulting in a final concentration of 5.9 ug/ml SAPE and 0.1% BSA per reaction. After a second incubation at 52°C for 15 minutes, 50 μl was analysed on a MAGPIX instrument (Luminex).

## Screening of a collection of pesticides

A collection of pesticides was obtained through the Innovative Vector Control Consortium (Liverpool, United Kingdom) and the MMV (Geneva, Switzerland). Compounds were first diluted in DMSO and then in human serum type A to a concentration 4 times above the final test concentration. Blood meals were prepared by mixing 40 μl of diluted compound with 40 μl of $4.10^6$ CFU/ml barcoded *Asaia* and 80 μl human type O red blood cells. Controls included vehicle (0.1% DMSO) and positive controls fipronil and deltamethrin, both at 10 μM. Blood meals were prepared in duplicate for each compound and transferred to 96-well plates in 2 different layouts (S2 Fig). *A. stephensi* mosquitoes were allowed to feed for 20 minutes and maintained at 26°C and 70% to 80% humidity. Moreover, 48 hours after feeding, live and dead mosquitoes were processed in separate pools as described above.

## Screening for malaria transmission-blocking compounds

Infectious *P. falciparum* gametocytes of parasite line NF54-HGL, expressing a GFP-luciferase fusion protein under control of the hsp70 promoter, were cultured in RPMI 1640 medium supplemented with 367 μM hypoxanthine, 25 mM HEPES, 25 mM sodium bicarbonate, and 10% human type A serum in a semiautomated system as previously described [20,60]. Furthermore, 72-μl aliquots of cultures containing mature stage V gametocytes were transferred to 96-well v-bottom plates (Corning Life Sciences, Amsterdam, the Netherlands) in duplicate in 2 different layouts (S2 Fig). Test compounds from the Pathogen Box (MMV, Geneva, Switzerland) were diluted in DMSO and then in RPMI 1640 medium supplemented with 10% human

serum type A, and 8 μl of diluted compound was added to the gametocytes in the plate to achieve a final compound concentration of 10 or 20 μM and a final DMSO concentration of 0.2%. Positive and negative controls included 10 μM and 0.2% DMSO, respectively. Plates were incubated at 37°C, 4% $CO_2$ and 3% $O_2$ for 24 hours in accordance with established methods for maintenance of infectious gametocytes [31,61]. Subsequently, plates were centrifuged briefly (750x$g$, 5′), and 70-μl supernatant was removed and replaced with 42.7 μl of heat inactivated human type A serum, 48-μl human type O red blood cells, and 5.3 μl of barcoded *Asaia* bacteria to a final density of $10^5$ CFU/ml. All procedures were performed at 37°C. Plates were then sealed and used for feeding to *A. stephensi* mosquitoes as described above. Following feeding, mosquitoes were maintained at 26°C and 70% to 80% humidity and starved for 2 days. From day 3 onwards, the mosquitoes were presented with cotton pads wetted in a 5% glucose solution supplemented with 100 μg/ml kanamycin twice a day for a duration of 2 hours each to minimise barcode cross-contamination through the glucose pads. Eight days after feeding, mosquitoes were harvested and homogenised in 96-well plates as described above. Infection status of individual mosquitoes was analysed by determining luciferase activity in 45 μl of the mosquito homogenate as described previously [62]. Background luminescence was determined by analysing 10 uninfected (unfed) mosquitoes. Mosquitoes were considered infected when the luminescence signal was greater than the mean + 5xσ of the signal in the negative control mosquitoes as described previously [20]. *Asaia* cultures from uninfected and infected mosquitoes were collected in separate pools for further analysis of barcode signals.

## Standard membrane feeding assays using glass feeders

Results from barcoded experiments were validated through standard membrane feeding assays using traditional glass feeders [20]. For testing for systemic insecticide activity, compounds were serially diluted in DMSO and then in DMEM medium and combined with human type A serum and type O red blood cells to achieve a final DMSO concentration of 0.1% in 40% haematocrit in a volume of 300 μl. Blood meals were placed in glass feeders warmed at 37°C and *A. stephensi* mosquitoes were allowed to feed for 15 minutes. Following feeding, nonfed mosquitoes were removed and the blood-fed mosquitoes were maintained at 26°C and 70% to 80% humidity for 48 hours. Subsequently, the number of live and dead mosquitoes was determined for each test condition. Testing for compound effects on transmission of *P. falciparum* gametocytes to *A. stephensi* mosquitoes was performed as described previously [20].

## Replicates and data analyses

To obtain sufficient numbers of fed mosquitoes, all test compounds were presented in replicate blood meals (S2 Fig). An average of 6 mosquitoes per blood meal was used in barcoded feeding experiments. With a 90% feeding efficiency, this resulted in approximately 10 fed mosquitoes per test condition. Mosquitoes were processed individually and rescued barcoded *Asaia* bacteria were pooled according to phenotype. Here, the *Asaia* from the replicate plates were combined for each phenotype. For each pool, barcode fragments were amplified and analysed in triplicate. Fluorescence intensity was determined by analyses of at least 40 microspheres per barcode and expressed as relative median fluorescence intensity (MFI). MFI values were averaged from the triplicates observations for each pool and corrected for average background signals from negative control (GLY medium without barcoded *Asaia*) samples. Barcodes were considered as sampled when the signal was above the mean + 3σ of the negative control samples. In comparative phenotypic analyses, data were expressed as the relative proportion of the barcode signal in the phenotype of interest. For example, when comparing uninfected and

infected mosquitoes, the percentage of the barcode signal in the uninfected mosquitoes was calculated by

$$P_u = 100 \times \left( \frac{I_u}{I_u + I_i} \right),$$

where $I_u$ and $I_i$ are the background corrected median fluorescence intensities in the uninfected and infected mosquitoes, respectively.

In standard membrane feeding experiments using glass feeders, all conditions were tested in 2 replicate feeders, and at least 24 mosquitoes were analysed per feeder. Data were analysed and visualised using the Prism software package (GraphPad Software, San Diego, US). IC$_{50}$ values for systemic insecticides were determined by fitting a 4 parameter logistic regression model using least squares to find the best fit. IC$_{50}$ values in *Plasmodium* transmission-blocking experiments were determined by assuming a beta binomial distribution and logistic regression using maximum likelihood to find the best fit as described previously [63]. Effects of *Pantoea* or *Asaia* on *P. falciparum* were analysed by ANOVA using a Kruskal–Wallis test and Dunn multiple comparison test.

## Supporting information

**S1 Movie. Feeding of Anopheles stephensi on blood meals in a 96-well microtiter plate.** The cage contained approximately 300 mosquitoes.
(M4V)

**S1 Fig.** (A) Device for stretching Parafilm in 2 directions. (B) Lever press used for applying the Parafilm membrane to a 96-well plate. (C) Feeding mosquitoes on a 96-well plate. The plate is heated by an aluminium heat block on top of the plate.
(TIFF)

**S2 Fig. Oligonucleotide pilot barcoding experiments and effects of symbiont bacteria on Plasmodium falciparum transmission.** (A) Pilot experiment with blood meals tagged with a phosphorothioate oligonucleotide. Mosquitoes were fed by membrane feeding on blood meals containing 0 to 10 ng/µl of oligonucleotide as indicated in the legend. At 0, 24, 72, and 144 hours postfeeding, mosquitoes were homogenised, total DNA was isolated, and the amount of oligonucleotide was determined by semiquantitative real time PCR. (B–E) Effect of Pantoea agglomerans (B and C) or Asaia SF2.1 (D and E) on transmission of P. falciparum NF54 parasites. The panels show data from independent experiments. Experiments were conducted with parasite strain NF54-HGL that expresses a GFP-luciferase reporter throughout the life cycle. Stage V gametocytes were combined with bacteria at the densities indicated on the x-axis and fed to Anopheles stephensi mosquitoes. Eight days after feeding, infection status was determined by luminescence analysis. The symbols indicate relative light units observed in individual mosquitoes. Asterisks indicate data significantly different from the control infection that received no bacteria (*P < 0.05, **P < 0.01;****P < 0.0001). Underlying data for this figure can be found in S1 Data.
(TIFF)

**S3 Fig. Proportion of barcode signal originating from live or dead mosquitoes for DMSO (upper panel) or fipronil barcoded blood meals.** Barcode signals were quantified from individual mosquitoes. For each barcode, all signals were summed. The figure shows the percentage of signal that was derived from dead versus live mosquitoes. Underlying data for this figure can be found in S1 Data.
(TIFF)

**S4 Fig. Plate layout used in phenotypic screening experiments.** All samples were tested in duplicate plates using the plate maps indicated in the figure.
(TIFF)

**S5 Fig. Number of compounds with systemic insecticide activity against Anopheles stephensi per IRAC chemical class.** Compounds were considered active when the associated barcode showed ≥50% enrichment in dead mosquitoes at one of the test concentrations (0.1; 1.0; 10.0 μM). Unclassified compounds or classes composed of <2 compounds are not included in the figure. Underlying data for this figure can be found in S1 Data.
(TIFF)

**S6 Fig. Outline of procedure for screening for malaria transmission-blocking compounds.** Test compounds were combined with infectious stage V gametocytes from Plasmodium falciparum strain NF54-HGL that expresses a luciferase reporter throughout the life cycle. Following 24-hour incubation, gametocytes were supplemented with uninfected red blood cells and barcoded Asaia bacteria and fed to Anopheles stephensi mosquitoes using 96-well microtiter plates. Eight days after blood feeding, infection status of individual mosquitoes was determined through luminescence assays. In parallel, Asaia bacteria were grown from homogenates of individual mosquitoes in 96-well liquid cultures under kanamycin selection pressure. Asaia were pooled according to infection status into pools for infected versus uninfected mosquitoes. Barcodes were then amplified by PCR using a fluorescently labelled primer pair that binds a common sequence flanking the DNA barcode sequence. Following amplification, barcodes were quantified by multianalyte profiling using DNA oligos coupled to colour-coded microspheres, which resulted in a fluorescence signal for each barcode depending on the quantity of the barcode in the PCR amplification product. Barcodes enriched in the uninfected mosquitoes identified compounds with malaria transmission-blocking activity, whereas detection of barcode signals from the infected mosquitoes were used to verify sampling of barcodes that were missing in the pool of uninfected mosquitoes.
(TIFF)

**S7 Fig. Composition of the Pathogen Box, a collection of compounds assembled by the MMV from hits of high-throughput screening campaigns from disease programmes as indicated in the legend.** Underlying data for this figure can be found in S1 Data. MMV, Medicines for Malaria Venture.
(TIFF)

**S8 Fig. Malaria transmission-blocking effects of selected compounds.** Experiments were conducted with P. falciparum strain NF54-HGL that expresses a luciferase reporter throughout the life cycle. Stage V gametocytes were preincubated with test compound for 24 hours prior to feeding to Anopheles stephensi mosquitoes. Eight days after feeding, infection status was determined by luminescence analysis. The symbols indicate relative oocyst intensities normalised to the luminescence signals observed in vehicle control (0.1% DMSO) infections. In line with the test concentrations in the barcoded screen, compounds MMV688754, MMV688978, MMV688122, MMV687248, MMV023969, MMV675993, MMV1088520, MMV688362, MMV022029, MMV687794, MMV687765, and MMV688122 were tested at 20 μM, all other compounds were tested at 10 μM. Underlying data for this figure can be found in S1 Data. MMV, Medicines for Malaria Venture.
(TIFF)

**S1 Table. Oligonucleotides used in pilot barcoding experiments and PCR amplifications.** Underlined nucleotides indicate phosphorothioate modifications.
(DOCX)

**S2 Table. Oligonucleotides used for generation of barcode plasmids.**
(DOCX)

**S3 Table. List of TAG regions on Luminex microspheres, primers used for barcode construction, and resulting barcoded plasmids.**
(DOCX)

**S4 Table. Compounds analysed in full dose–response in the SMFA.** Plasmodium asexual blood stage and gametocyte inhibition data were retrieved from https://www.mmv.org/mmv-open/pathogen-box. SMFA, standard membrane feeding assay.
(DOCX)

**S1 Data. Underlying data of figures.**
(XLSX)

## Acknowledgments

We wish to thank Claudia Damiani and Aida Capone for help with the *Asaia* SF2.1 strain, Marcelo Jacobs-Lorena for his kind gift of *Pantoea agglomerans*, and Sander Koenraadt for provision of *Aedes aegypti* mosquitoes. Bernd Engelbrecht, Katharina Schumacher, and colleagues at Irmato Industrial Solutions are gratefully acknowledged for help with the design of the plate sealing process. The authors wish to thank Geert-Jan van Gemert and Laura Pelsen-Posthumus for expert technical assistance in mosquito rearing. Sarah Rees is acknowledged for provision of a collection of pesticides. The authors thank Isaac Sandoval Capuchino for providing artwork. Manuel Llinás and Robert Sauerwein are gratefully acknowledged for critical reading of the manuscript.

## Author Contributions

**Conceptualization:** Angelika Sturm, Kirandeep Samby, Esperanza Herreros, Koen J. Dechering.

**Formal analysis:** Angelika Sturm, Martijn W. Vos.

**Funding acquisition:** Koen J. Dechering.

**Investigation:** Angelika Sturm, Martijn W. Vos, Rob Henderson, Maarten Eldering, Karin M. J. Koolen, Avinash Sheshachalam.

**Methodology:** Angelika Sturm, Martijn W. Vos, Maarten Eldering, Guido Favia.

**Project administration:** Angelika Sturm, Esperanza Herreros, Koen J. Dechering.

**Resources:** Guido Favia, Kirandeep Samby.

**Supervision:** Angelika Sturm, Esperanza Herreros, Koen J. Dechering.

**Visualization:** Angelika Sturm.

**Writing – original draft:** Angelika Sturm, Martijn W. Vos.

**Writing – review & editing:** Guido Favia, Kirandeep Samby, Esperanza Herreros, Koen J. Dechering.

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
