## [Editor Report · Decision Letter 0]

15 Sep 2021

Dear Dr. Dechering, 

Thank you for submitting your manuscript entitled "The use of barcoded Asaia bacteria in mosquito in vivo screens for identification of inhibitors of malaria transmission" for consideration as a Research Article by PLOS Biology.

Your manuscript has now been evaluated by the PLOS Biology editorial staff and I am writing to let you know that we would like to send your submission out for external peer review.

Please note, however, that the outcome of our discussion of your manuscript is that we have some reservations as to the depth of analysis and the overall novelty offered by your system. We would need to be persuaded by the reviewers that the paper has the potential after revision to offer the significant strength of advance that we require for publication in order to pursue it further for PLOS Biology.

Before we can send your manuscript to reviewers, we need you to complete your submission by providing the metadata that is required for full assessment. To this end, please login to Editorial Manager where you will find the paper in the 'Submissions Needing Revisions' folder on your homepage. Please click 'Revise Submission' from the Action Links and complete all additional questions in the submission questionnaire. We will consider your manuscript as a Methods and Resources manuscript, so please choose that option where corresponds in the system when resubmitting.

Please re-submit your manuscript within two working days, i.e. by Sep 17 2021 11:59PM.

Kind regards,

Paula 

---

Paula Jauregui, PhD

Associate Editor

PLOS Biology

---

## [Decision Letter · Decision Letter 1]

30 Oct 2021

Dear Dr. Dechering,

Thank you for submitting your manuscript "The use of barcoded Asaia bacteria in mosquito in vivo screens for identification of systemic insecticides and inhibitors of malaria transmission" for consideration as a Methods and Resources at PLOS Biology. Your manuscript has been evaluated by the PLOS Biology editors, an Academic Editor with relevant expertise, and by several independent reviewers.

In light of the reviews (below), we are pleased to offer you the opportunity to address the comments from the reviewers in a revised version that we anticipate should not take you very long. We will then assess your revised manuscript and your response to the reviewers' comments and we may consult the reviewers again.

We suggest you change the title to make it more accessible. This is our suggestion: "Barcoded Asaia bacteria enable mosquito in vivo screening and identify novel systemic insecticides and inhibitors of malaria transmission".

Please also address the following editorial and policy requests:

DATA POLICY:

Regardless of the method selected, please ensure that you provide the individual numerical values that underlie the summary data displayed in the following figure panels as they are essential for readers to assess your analysis and to reproduce it: Figures 1ABC, 2ABC, 3ABCD, 4ABCE, S2ABCD, S3, S5, S7, S8.

We expect to receive your revised manuscript within 1 month.

**IMPORTANT - SUBMITTING YOUR REVISION**

*Resubmission Checklist*

*Published Peer Review*

*PLOS Data Policy*

*Blot and Gel Data Policy*

Sincerely,

Paula

---

Paula Jauregui, PhD

Associate Editor

PLOS Biology

REVIEWS:

Reviewer #1: Paul M. Selzer. Industrial research on small molecules for control of ecto- and endoparasites

Reviewer #2: Malaria transmission and drug-based vector control.

Reviewer #1: PBIOLOGY-D-21-02346

The use of barcoded Asaia bacteria in mosquito in vivo screens for identification of inhibitors of malaria transmission

The authors describe a new methodology which can be used to aid the discovery of novel insecticides. Genetically engineered alpha-proteobacteria (Asaia) containing a DNA-barcode was added to bloodmeals in combination with compounds to be tested. A custom-built device was developed to facilitate the feeding protocol. The mosquitoes are pooled according to phenotype and the bacteria grown. The DNA barcodes corresponding to different phenotypes are then amplified and quantified. Barcodes enriched in the dead mosquitoes identified compounds with systemic adulticidal activity. In a subsequent series of experiments this system also allowed for compounds to be tested for their efficacy in targeting sexual stages of the parasite by employing P. falciparum gametocytes expressing a GFP-luciferase fusion protein for detection. By killing the mosquito vector or targeting the sexual stages of the parasite, the transmission of malaria would be interrupted. 

The manuscript is well written, the methodology is innovative, well explained and might represent a game-changer in screening compounds acting as insecticides or targeting the sexual stages of P. falciparum. The process might even be adapted to other parasitic species which depend on blood-feeding vectors.

Comments:

Figure 1b - Why was the uptake/fluorescence of barcode B so much lower in some mosquitoes when compared to barcode A?

Figure 4D - "The lower left corner of each panel indicates the proportion of barcode signal that was retrieved from uninfected mosquitoes". How was this proportion calculated? Is it a percentage?

"Compounds identified by a blue dot were selected for full dose response analysis". Why were these compounds specifically chosen?

Line 235 - Is the number 23 a reference?

Line 436 - If plates containing infectious P. falciparum gametocytes of the parasite were incubated for 24h with compounds, would that not generate false positives? The compounds have a 24h window to kill the gametocytes before the addition of the bacteria and before the feeding assays, in an environment which does not reflect their natural habitat.

Reviewer #2: This is a nicely written paper that cleverly repositions some older ideas into an effective and efficient screen for testing systemic insecticides and Plasmodium transmission blocking (TB) molecules in mosquito blood meal bioassays. This is very novel, important not just for the field but could broadly facilitate discovery of biomolecules that help us control many different mosquito borne diseases, and has already shown effectiveness in drug discovery in the case of the TB assays given the several new effective compounds discovered from the MMV Pathogens Box. Overall, I believe their methods do significantly outperform their predecessors in precision, resolution, speed and accessibility. I think their controls are well done, and prove the efficiency of their tests. I like that they have done a good job of showing that mosquitoes can and do sample multiple wells, but that their barcodes and plate set up allows for discrimination of this and so they can handle the false positives and false negatives that happen from this habit. I think it is very clever that their serial testing adds in lower or higher dose testing of the compounds to verify positive hits by re-testing at a lower dose, and screens for false negatives by re-testing at a higher dose. I think they did their due diligence on their drug hits by going to the standard, repeat membrane feeds using group membrane feeds at serial doses. I have several broad critiques and questions about the present manuscript that I hope the authors can answer.

1) The novelty of their idea and technology is reconfiguring some older/other ideas and adding a few new ones to make an efficient screen technology for blood feeding mosquitoes. That said, they should do a better job of describing those foundational ideas, particularly the idea of barcoding mosquitoes/blood meals. 

a. The Killeen et al work, buy virtue of using phage display, did not just develop the idea of blood feeding Anopheles on 96 well plates, but also 'bar-coded' the wells by virtue of testing one scFv-displaying phage per well, which contained the genetic sequence of that particular scFv in the phage genome, and phages that had supposed activity were recovered from the guts of dead mosquitoes, re-amplified and re-tested. Ghosh et al (PMID 11687659) did a similar thing in 2001, but did not use the multi-well idea.

b. Bar-coding individual/small groups of mosquitoes for better assay discrimination is also being done a lot now in various ways. For example, it is being used in mark-release-recapture experiments in the field to discriminate different pools of mosquitoes that are released at different times (Faimen et al - https://www.biorxiv.org/content/10.1101/2020.08.23.262741v1)), 

c. Ideas like the Vectorchip (https://www.biorxiv.org/content/10.1101/2020.10.19.345603v1.full) are testing individual mosquitoes blood feeding on individual wells to discriminate mosquito transmission among a group.

2) After reading the whole manuscript, I understood the need to use Asaia for the bar coding - because it colonizes the gut and so allows for detecting the bar code 8 days post blood feeding when the TB assays for oocyst infection needs to be done. However, I was confused early on when the focus was only on systemic mosquito insecticides because I didn't understand why Asaia was needed when the mortality/live screen was only done 2 days post blood meal. To my mind, such a short timeframe would have allowed the investigators to simply put in a genetic barcode oligo or plasmid directly in the wells, and simply do PCR of the guts of the live vs. dead mosquitoes because 2 days is not enough time to digest these in the midgut; adding the barcoded Asaia seems to add an unnecessary culture step that slows the screen. Can the authors explain this? Was the idea just to keep the steps the same between the two types of tests? Would the mortality screen indeed be effective and/go faster if what I described is changed for the mortality screen only?

3) Similar to the above comment, I am wondering about the use of MagPix for the output of determining bar code relative intensity. Again, it seems it may be needed for the oocyst detections because oocyst number/intensity is the readout, but for a live/dead screen, don't they just need a +/- determination of their assay (PCR or qPCR)…which would make it easier and more efficient because no Asaia culture, nor an expensive MagPix machine and fluorescent beads, is needed? For example, in Fig2 - if the y-axis is a binary determination on whether fipronil (the pos control) or DMSO (the neg control) was sampled, the more complicated bar code relative intensity readout is not really necessary, right? 

4) Line 185, they refer to the idea that interference from a gustatory effect was counted out in their testing. This was a good test of screening out artifacts, but actually testing FOR gustatory effects would be interesting and potentially useful in control strategies. Can the authors comment on that?

5) Fig 3B shows negative and >100% signals in dead mosquitoes, which is counter-intuitive. I think this is from the idea of multiple wells being sampled by mosquitoes but I'm not sure. Please explain what is going on here.

6) For the TB effects, the readout is RLU of mosquitoes from oocyst intensity. This is an efficient way to do this to prevent the need to dissect and count oocysts, but it also prevents us from understanding the RLU-to-oocyst number relationship. Please add data that gives us this understanding. Only 1 oocyst is needed to transmit the parasite. If a drug reduces RLU intensity by 50%, it may be statistically technically significant but may be irrelevant in the ability to reduce malaria transmission using the drug (e.g going from 10 down to 5 oocysts is unlikely to affect malaria transmission by the mosquito). This should be acknowledged, and the effectiveness of the positive hits couched in this critique - changing oocyst intensity is only really going to matter if we change it to zero from what ever number.

7) Fig 4e shows a legend for Rep 1 and Rep 2 but with no distinguishment between the two (eg. different colored dots). I wonder if the authors meant to color the individual dots that construct the SMFA dose response analysis by Rep 1 vs. Rep 2?....this would be helpful.

8) Line 319 - they should briefly explain the rule-of-five principles.

9) Could the authors expound upon the idea that their test could allows for screening of slow killing systemic insecticides, and for screening for egg-laying inhibitors? Both assays would be very useful - the former potentially get hits for chemicals that would potentially be more resistant to insecticide resistance-development and the latter help to reduce the reservoir population. 

10) I don't understand what the authors are saying in sentences 392-395. Please revise.

11) Line 452 - explain "cherry-picked"

---

## [Editor Report · Decision Letter 2]

3 Dec 2021

Dear Dr. Dechering,

On behalf of my colleagues and the Academic Editor, Luis Teixeira, I am pleased to say that we can in principle accept your Methods and Resources paper "Barcoded Asaia bacteria enable mosquito in vivo screens and identify novel systemic insecticides and inhibitors of malaria transmission" for publication in PLOS Biology, provided you address any remaining formatting and reporting issues. These will be detailed in an email that will follow this letter and that you will usually receive within 2-3 business days, during which time no action is required from you. Please note that we will not be able to formally accept your manuscript and schedule it for publication until you have any requested changes.

PRESS

Sincerely, 

Paula

---

Paula Jauregui, PhD 

Associate Editor 

PLOS Biology
